# Evaluating the use of the QUiPP app and its impact on the management of threatened preterm labour: A cluster randomised trial

**Helena A. Watson** ⓘ *, **Naomi Carlisle** ⓘ, **Paul T. Seed** ⓘ, **Jenny Carter** ⓘ, **Katy Kuhrt** ⓘ, **Rachel M. Tribe** ⓘ, **Andrew H. Shennan** ⓘ

Department of Women and Children's Health, School of Life Course Sciences, King's College London, St Thomas' Hospital, London

* helenawatson85@gmail.com

**Data Availability Statement:** All relevant data are within the manuscript and its Supporting Information files.

## Abstract

### Background

Preterm delivery (before 37 weeks of gestation) is the single most important contributor to neonatal death and morbidity, with lifelong repercussions. However, the majority of women who present with preterm labour (PTL) symptoms do not deliver imminently. Accurate prediction of PTL is needed in order ensure correct management of those most at risk of preterm birth (PTB) and to prevent the maternal and fetal risks incurred by unnecessary interventions given to the majority. The QUantitative Innovation in Predicting Preterm birth (QUIPP) app aims to support clinical decision-making about women in threatened preterm labour (TPTL) by combining quantitative fetal fibronectin (qfFN) values, cervical length (CL), and significant PTB risk factors to create an individualised percentage risk of delivery.

### Methods and findings

EQUIPTT was a multi-centre cluster randomised controlled trial (RCT) involving 13 maternity units in South and Eastern England (United Kingdom) between March 2018 and February 2019. Pregnant women ($n$ = 1,872) between $23^{+0}$ and $34^{+6}$ weeks' gestation with symptoms of PTL in the analysis period were assigned to either the intervention (762) or control (1,111). The mean age of the study population was 30.2 (+/− SD 5.93). A total of 56.0% were white, 19.6% were black, 14.2% were Asian, and 10.2% were of other ethnicities. The intervention was the use of the QUiPP app with admission, antenatal corticosteroids (ACSs), and transfer advised for women with a QUiPP risk of delivery >5% within 7 days. Control sites continued with their conventional management of TPTL. Unnecessary management for TPTL was a composite primary outcome defined by the sum of unnecessary admission decisions (admitted and delivery interval >7 days or not admitted and delivery interval ≤7 days) and the number of unnecessary in utero transfer (IUT) decisions/ actions (IUT that occurred or were attempted >7 days prior to delivery) and ex utero transfers (EUTs) that should have been in utero (attempted and not attempted). Unnecessary management of TPTL was 11.3% (84/741) at the intervention sites versus 11.5% (126/

**Funding:** The development of the QUIPP app and the EQUIPTT study are funded by the Guy's and St Thomas' Charity (Registered Charity No. 1160316) and Tommy's (1060508). The funders had no role in study design, data collection and analysis, decision to publish, or preparation of the manuscript.

**Competing interests:** I have read the journal's policy and the authors of this manuscript have the following competing interests. AS is Principal Investigator on Hologic funded science grants, which are paid directly to institute. NC received financial assistance to cover expenses only, paid directly to institution, to provide educational talks and an article on preterm birth from Hologic, USA. The other authors report no conflicts of interest.

**Abbreviations:** ACS, antenatal corticosteroid; CI, confidence interval; CL, cervical length; CONSORT, Consolidated Standards of Reporting Trials; EUT, ex utero transfer; GCP, Good Clinical Practice; IUT, in utero transfer; NHS, National Health Service; NICE, National Institute for Health and Care Excellence; NICU, neonatal intensive care unit; OR, odds ratio; PCN, Preterm Clinical Network; PPROM, preterm premature rupture of the membranes; PTB, preterm birth; PTL, preterm labour; qfFN, quantitative fetal fibronectin; QUIPP, QUantitative Innovation in Predicting Preterm birth; RCT, randomised controlled trial; ROC, receiver operating characteristic; sPTB, spontaneous preterm birth; TPTL, threatened preterm labour.

1094) at control sites (odds ratio [OR] 0.97, 95% confidence interval [CI] 0.66–1.42, $p = 0.883$). Control sites frequently used qfFN and did not follow UK national guidance, which recommends routine treatment below 30 weeks without testing. Unnecessary management largely consisted of unnecessary admissions which were similar at intervention and control sites (10.7% versus 10.8% of all visits). In terms of adverse outcomes for women in TPTL <36 weeks, 4 women from the intervention sites and 12 from the control sites did not receive recommended management. If the QUiPP percentage risk was used as per protocol, unnecessary management would have been 7.4% (43/578) versus 9.9% (134/1,351) (OR 0.72, 95% CI 0.45–1.16). Our external validation of the QUiPP app confirmed that it was highly predictive of delivery in 7 days; receiver operating curve area was 0.90 (95% CI 0.85–0.95) for symptomatic women. Study limitations included a lack of compliance with national guidance at the control sites and difficulties in implementation of the QUiPP app.

## Conclusions

This cluster randomised trial did not demonstrate that the use of the QUiPP app reduced unnecessary management of TPTL compared to current management but would safely improve the management recommended by the National Institute for Health and Care Excellence (NICE). Interpretation of qfFN, with or without the QUiPP app, is a safe and accurate method for identifying women most likely to benefit from PTL interventions.

## Trial registration

ISRCTN Registry ISRCTN17846337.

---

## Author summary

### Why was this study done?

- Accurate diagnosis of premature labour is desirable in order ensure optimum management of those most at risk of preterm birth (PTB) and to prevent the maternal and fetal risks incurred by unnecessary interventions to the majority of women who do not deliver within a week of presentation.

### What did the researchers do and find?

- The QUantitative Innovation in Predicting Preterm birth (QUIPP) app is a clinical decision-making aid based on woman's individual risk factors for premature birth and quantitative fetal fibronectin (qfFN) values and/or cervical length (CL) as determined by transvaginal ultrasound, both of which are existing point-of-care tests.

- This trial randomised maternity units to use the QUiPP app to triage women with symptoms of premature labour versus the unit's conventional management.

- The study also provided a dataset to test the accuracy of the QUiPP app on a population other than that on which it was built.

- The QUiPP intervention did not succeed in lowering unnecessary admission and transfer decisions relative to the control sites.

- The trial provided evidence that the QUiPP is accurate and safe: No women missed timely treatments with its use.

**What do these findings mean?**

- The trial did not provide evidence that the QUiPP app reduced unnecessary treatments compared to routine management.

- The authors believe that this was due to the larger than anticipated noncompliance with national guidance which does not recommend predictive tests below 30 weeks' gestation and problems in ensuring all clinicians used the app at interventions sites.

## Introduction

Symptoms suggestive of preterm labour (PTL) are one of the most common reasons for mothers presenting to hospital antenatally, although very few will deliver imminently [1–4]. Accurate prediction of PTL is needed in order to ensure correct management of those most at risk of preterm birth (PTB) and to prevent the maternal and fetal risks incurred by unnecessary interventions given to the majority of women who do not deliver within a week of presentation [5–7]. Clinical intervention "just in case" a woman may deliver early results in many women being transferred out of their local hospital unnecessarily and receiving unwarranted drugs, such as steroids [8]. It also prevents necessary transfers as neonatal cots are blocked needlessly, resulting in more potentially hazardous ex utero transfers (EUTs) [9].

An increase in cervicovaginal quantitative fetal fibronectin (qfFN) and cervical length (CL) shortening are likely to represent a common pathway for pathological activation of labour and have the broadest evidence base [10–12]. Our research group developed the "QUantitative Innovation in Predicting Preterm birth" (QUIPP) app, which is a clinical decision support tool based on qfFN values, CL, and risk factors of women with symptoms or those considered to be at high risk of PTB, which is an accurate indicator of PTB risk [4,13–16].

While the QUIPP app has been shown to be accurate and downloaded and used widely by clinicians, it requires formal evaluation in routine clinical practice. Here, we present the results of a multisite cluster randomised controlled trial (RCT) to evaluate the use of the QUiPP app and its subsequent impact on reducing unnecessary management for threatened preterm labour (TPTL). In addition, this study allowed for an external validation of the QUIPP app prediction models.

## Methods

### Aims and objectives

The primary aim of the EQUIPTT study was to evaluate the use of the QUIPP app for management of TPTL and potential reduction of unnecessary management.

## Hypothesis

The implementation of the QUIPP app and management algorithm will decrease unnecessary management for TPTL (following the current National Institute for Health and Care Excellence [NICE] guidance).

A secondary aim was to externally validate the QUIPP app prediction models in an emergency obstetric setting.

## Trial design and participants

EQUIPTT (REC reference 17/LO/1802) was a cluster RCT (with a parallel group design) across 13 obstetric centres. During the intervention phase, each maternity unit required capacity for quantitative assessment of qfFN and/or transvaginal ultrasonic CL measurement. A pragmatic approach was taken to introduce the intervention (i.e., use of the QUIPP app and related TPTL management guidance) to the entire hospital antenatal units as a standard practice for all clinicians and application to all affected pregnancies. However, since individuals in the same cluster tended to have more similar outcomes than those across clusters, an intra-class correlation coefficient was used, creating a larger sample size than would be required for an individual RCT.

## Randomisation

Randomisation was performed at the cluster (maternity centre) level. A computer-generated random allocation sequence was used, stratified by the level of special care available. Randomisation was designed to allocate approximately equal numbers to each arm according to the level of care provided: neonatal intensive care unit (NICU) versus local neonatal unit or special care baby unit. Due to their being an odd number of centres, 7 sites were allocated to one group and 6 to the other.

## Procedures

All 13 centres provided data related to the primary outcome under their current standard practice in a 6-week pre-intervention data collection period. The baseline data obtained described existing practices for triage of TPTL and were used for adjusting effects due to differences between clusters. Following randomisation, the centres were instructed to either implement use of the QUiPP app and management guidance (intervention) or to follow their current standard management (control) for a 9-month analysis period (March 26, 2018 to December 31, 2019). It was anticipated that this would be similar to current NICE guidance for the management of PTL. In the final phase of the trial, the intervention was introduced in the 6 control units, and data were collected to investigate the impact of the intervention in control sites.

Eligible women were between $23^{+0}$ and $34^{+6}$ weeks of pregnancy presenting to labour ward or day assessment units with symptoms of TPTL (such as contractions or abdominal pain). Exclusion criteria were a definitive diagnosis of labour (i.e., regular painful contractions with cervical change >3 cm on speculum or digital examination), confirmed ruptured membranes (on speculum examination), or significant vaginal bleeding.

As with many cluster trials, since the QUIPP intervention was an educational decision-making tool rolled out once to all clinicians, implementation was at the institutional level, and individual consent was not appropriate or logistical. The no-consent model was specifically addressed in the ethics application, and the ethics committee waived the need for informed consent (REC reference 17/LO/1802).

All intervention sites received face-to-face training on the use of the QUIPP app and management guidance prior to its introduction. Additional support (either face to face or via email/telephone) was also available throughout the trial. To use the QUiPP app, the doctor/midwife assessing the woman needed to input the gestation, previous history of late miscarriage or spontaneous preterm birth (sPTB), qfFN value, and/or ultrasonic CL into the app. The output of the app then provided a % risk of delivery within 1, 2, and 4 weeks and before 30, 34, and 37 weeks. The exact QUiPP % thresholds for admission or treatment are gestation dependent, and, therefore, could be tailored to individual circumstances. However, the study guidance suggested a 5% risk of delivery within 7 days as a threshold for antenatal corticosteroid (ACS) administration, admission, and/or in utero transfer (IUT). A 5% threshold was chosen based on the minimum level regarded as warranting intervention in our Delphi consensus [17] and the previous definitions of a low PTB risk provided by other preterm prediction studies [18–19].

Only anonymised data were stored on the secure internet-based Preterm Clinical Network (PCN) Database (MedSciNet) (REC reference 16/ES/0093). All investigators and study site staff complied with the requirements of the Data Protection Act 1998 with regard to the collection, storage, processing, and disclosure of personal information and upheld the Act's core principles. Further details of trial recruitment and governance procedures are available in the published trial protocol [20].

## Outcome measures

The primary outcome of unnecessary management for TPTL was defined by the number of unnecessary admission decisions: admitted and delivery interval >7 days or not admitted and delivery interval ≤7 days, and the number of unnecessary IUT decisions/actions: IUTs that occurred or were attempted >7 days prior to delivery and EUTs that should have been in utero (attempted and non-attempted). EUTs that should have been in utero were defined as babies transferred within 24 hours of birth.

Secondary outcomes included all individual components of primary outcome, maternal clinical outcomes (e.g., new onset gestational diabetes, thromboembolic disease, and confirmed sepsis), neonatal clinical outcomes (e.g., neonatal death prior to discharge, gestational age at delivery, birth weight, and days of supplemental oxygen), process measures (days of maternal hospitalisation, steroid, tocolytic and magnesium sulphate administration, and NICU admissions), and compliance with management recommendations.

## Statistical analysis

Data analysis followed the intention to treat principle, according to the planned intervention. Data were analysed using Stata Version 15 software or later (Stata, College Station, Texas, United States of America) to estimate the size and test for statistical significance of any effects of the intervention on primary and secondary outcomes. While our protocol planned to express treatment effects as risk ratios (relative risk), at analysis, we opted for a multilevel model in Stata, which provides odds ratios (ORs) with 95% confidence intervals (CIs), using binomial regression and adjusting for variables used in the minimisation process. Risk differences were also calculated for the primary outcome. The analysis model included a random effect for centre and standard errors adjusted for clustering by centre. Adjustments were also made for differences between cluster populations, such as ethnicity and maternal age. Our primary outcome of unnecessary decisions was planned to be measured per 1,000 deliveries. Unnecessary management was calculated per woman rather than per visit: a woman with any visit which resulted in unnecessary care being counted as having experienced unnecessary

management. This approach avoided the clustering effects of counting all visits and the problems of selecting a single visit to represent the care for one woman.

For unnecessary admission decisions, a total sample size of 580 was calculated; this equated to approximately 50 recruits per site, based on 12 sites (13th site added after power calculation performed). Data from our group's prospective observational study into the ability of qfFN to predict sPTB in symptomatic women (PETRA REC reference 14/LO/1988) allowed us to estimate the likely treatment effect for the intervention reducing unnecessary admissions from 25% to 10% and intra-class correlation coefficient of 0.035. This treatment effect required 580 for 90% power. We were aware that the statistical power of EQUIPTT could be enhanced by more clusters, but for the present study, 12 to 14 centres was considered the limit of what was feasible [21] considering the parallel cluster design.

We anticipated that the cluster design's randomisation of the QUiPP intervention at the institutional level relies on high uptake and adoption of the intervention, which is not a problem for control sites continuing usual care. To estimate the contribution of this effect, we also planned a per-protocol analysis to evaluate how the primary outcome would be impacted by ideal use of the QUiPP app, i.e., on all eligible episodes of TPTL and with adherence to the 5% threshold to guide management decisions. We adhered to the Consolidated Standards of Reporting Trials (CONSORT) checklist (S1 CONSORT Checklist) for conduct and reporting of this trial.

We have previously described the creation of the prognostic models for the QUiPP app v.2 symptomatic algorithms, the simple calibration of these models, and a temporal internal validation. Using % risk of ≥5% as an indication of a positive test, the area under the curve for prediction of PTB within 7 days using the qfFN algorithm was 0.893 [4]. However, external validation was required to test the generalisability of the prognostic models and correct for overfitting. To calculate the performance of the QUiPP app symptomatic qfFN predictive model, a validation set was created from EQUIPTT participants with a qfFN value from the duration of the trial period across all 13 sites. Any qfFN values ($n = 1$) documented as >500 were changed to 500 (as the qfFN machine will not provide a value >500 ng/mL).

## Ethics approval

The trial was conducted in compliance with the principles of the Declaration of Helsinki (1996), the principles of Good Clinical Practice (GCP), and in accordance with all applicable regulatory requirements including but not limited to the Research Governance Framework and the Medicines for Human Use (Clinical Trial) Regulations 2004, as amended in 2006 and any subsequent amendments. EQUIPTT was granted a favourable ethical opinion (REC reference 17/LO/1802) by the London Bridge Research Ethics Committee on November 21, 2017.

## Results

During the trial period, after exclusions (Fig 1), data from 243 women with 2,847 hospital visits were eligible for inclusion. Data errors and inconsistencies were identified by the trial team contemporaneously, and clarifications were sought from the recruiting site and corrected locally. Further data checks were performed by the trial statistician after the data were imported into Stata Version 15, for pre-analysis checks.

Table 1 describes study cluster characteristics and demographics as well as PTB risk factors of 1,872 recruited women in cluster analysis period (761 intervention and 1,111 control sites).

During the entire trial, there were 2,326 women with sufficient primary outcomes for analysis and 1,799 women in the cluster analysis period, 724 women at intervention sites, and 1,075 women at control sites. The lost to follow-up rate was 5% (37/761) in intervention sites and 3%

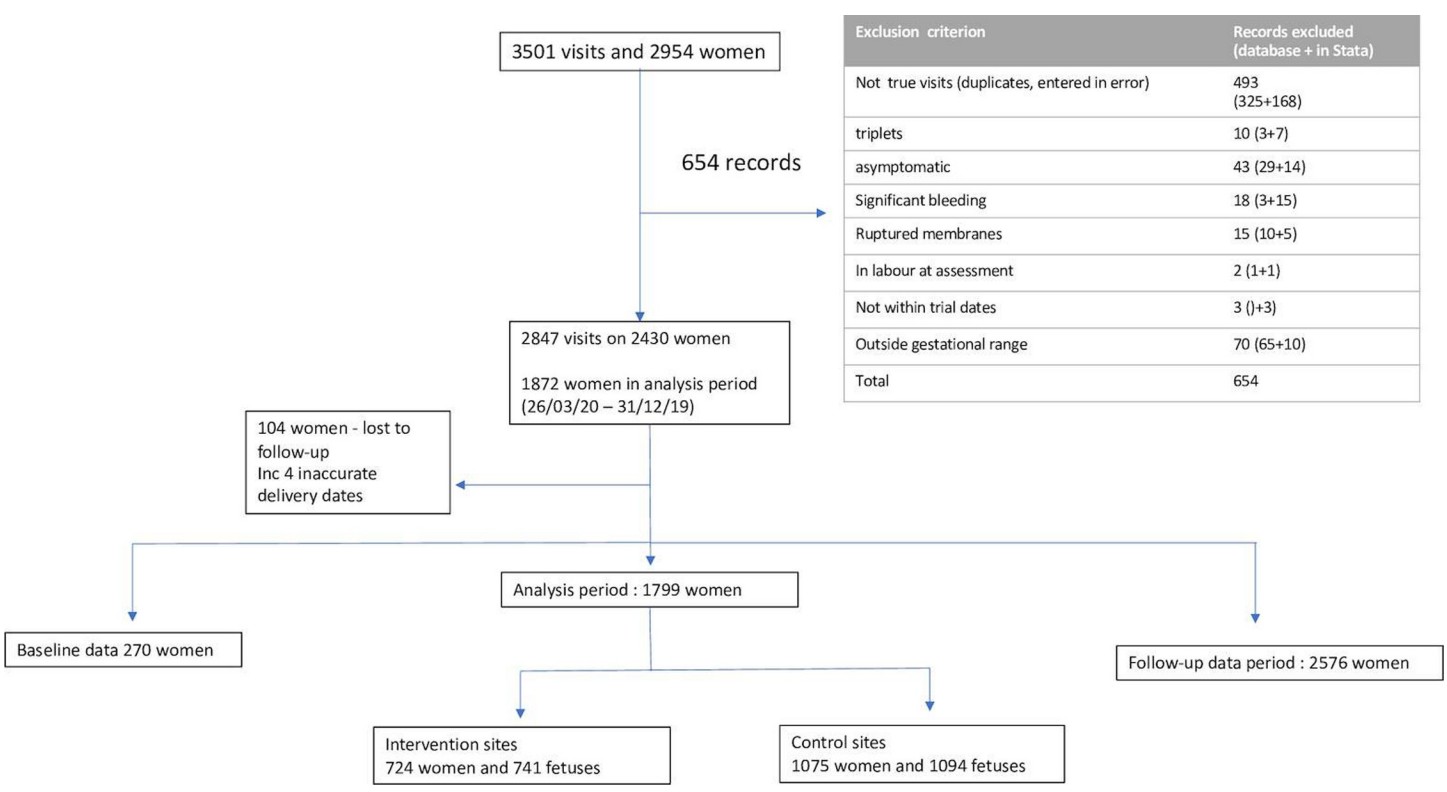

| Exclusion criterion | Records excluded (database + in Stata) |
|---|---|
| Not true visits (duplicates, entered in error) | 493 (325+168) |
| triplets | 10 (3+7) |
| asymptomatic | 43 (29+14) |
| Significant bleeding | 18 (3+15) |
| Ruptured membranes | 15 (10+5) |
| In labour at assessment | 2 (1+1) |
| Not within trial dates | 3 ()+3) |
| Outside gestational range | 70 (65+10) |
| Total | 654 |

**Fig 1. Flowchart to describe selection of women for EQUIPTT trial analysis.**

(36/1,111) in control sites. Unnecessary management of TPTL was 11.3% at the intervention sites versus 11.5% at control sites (OR 0.972, 95% CI 0.66 to 1.42). Unnecessary management largely consisted of unnecessary admissions that did not appear to be impacted by the intervention (10.7% versus 10.8% of all visits resulted in unnecessary admissions) (Table 2). The proportion of all admissions that were unnecessary was 43.8% (67/153) at intervention sites and 42.6% (84/197) at control sites.

There were 1,794 neonatal outcomes available with sufficient data for analysis of secondary outcomes (Table 3). ACS were defined as appropriate if at least 1 dose was administered ≥24 hours and ≤7 days before a delivery which occurred <36 weeks. ACS was classified as unnecessary if there was >7 days between time first dose was administered and delivery.

As per our prespecified per-protocol analysis, the proportion of unnecessary admissions and discharges was calculated for women as if the 5% threshold was always followed with the given QUiPP risks. Since 192 (27%) of women in the intervention arm did not have a QUiPP risk calculated at every visit, this reduced the number of women in the intervention arm with sufficient data for analysis to 519 across 761 visits. As for the primary outcome, we adjusted for baseline levels of unnecessary management using mixed effects logistic regression to account for errors between clusters and women. As described in Table 4, unnecessary admissions and discharges would have been reduced in the intervention arm if the QUiPP risk was used as per protocol (7.4% versus 9.9%), but this did not reach statistical significance.

## Adverse events

No "serious unexpected adverse event" (deliveries less than 30 weeks' gestation which occur outside of hospital) as described in the trial protocol [20] occurred during the trial period at

**Table 1. Baseline characteristics of clusters and women in intervention sites using QUiPP app versus control sites (n = 1,872).**

| Characteristic | Intervention | Control | Combined |
|---|---|---|---|
| *Cluster level* | | | |
| Number of hospitals, *n* | 6 | 7 | 13 |
| Maternity service *n* (%) <br> • Secondary <br> • Tertiary | <br> 867 (79.2) <br> 228 (20.8) | <br> 938 (55.0) <br> 768 (45.0) | <br> 1,805 (64.4) <br> 996 (35.6) |
| Neonatal service, *n* (%) <br> • Special care unit (level 1) <br> • Local neonatal unit (level 2) <br> • NICU (level 3) | <br> 617 (56.3) <br> 198 (18.1) <br> 280 (25.6) | <br> 938 (55.0) <br> 0 (0.0%) <br> 768 (45.0) | <br> 1,555 (55.5) <br> 198 (7.1) <br> 1,048 (37.4) |
| *Individual level* | | | |
| Total *n* | 761 | 1,111 | 1,872 |
| Mean maternal age at first visit, years, (± SD) | 29.61 (5.84) (*n* = 754) | 30.67 (6.02) (*n* = 1,087) | 30.24 (5.97) (*n* = 1,841) |
| Primiparity *n*/total (%) | 312/745 (41.9) | 485/1,098 (44.2) | 797/1,843 (43.2) |
| Mean body mass index at booking, kg/m$^2$ (± SD) | 26.29 (6.29) (*n* = 572) | 25.73 (5.72) (*n* = 1,050) | 25.93 (5.93) (*n* = 1,622) |
| Ethnicity, *n* (%) <br> • Asian (including Chinese) <br> • Black <br> • White <br> • Other | *n* = 661 <br> 126 (19.1) <br> 96 (14.5) <br> 376 (56.9) <br> 63 (9.5) | *n* = 1,013 <br> 112 (11.1) <br> 232 (22.9) <br> 561 (55.4) <br> 112 (11.1) | *n* = 1,674 <br> 238 (14.2) <br> 328 (19.6) <br> 937 (56.0) <br> 171 (10.2) |
| Smoking, *n* (%) <br> • Never <br> • Ex: gave up before pregnancy <br> • Ex: gave up in pregnancy <br> • Current smoker | *n* = 690 <br> 454 (65.8) <br> 85 (12.3) <br> 47 (6.8) <br> 104 (15.1) | *n* = 1,038 <br> 810 (78.0) <br> 89 (8.6) <br> 45 (4.3) <br> 93 (9.0) | *n* = 1,728 <br> 1,264 (73.1) <br> 174 (10.1) <br> 92 (5.3) <br> 197 (11.4) |
| Previous sPTB (<37$^{+0}$), *n*/total (%) | 81/758 (10.7) | 106/1,107 (9.6) | 187/1,865 (10.0) |
| Previous preterm prelabour rupture of membranes, *n*/total (%) | 23/758 (3.0) | 16/1,107 (1.4) | 39/1,865 (2.1) |
| Previous late miscarriage (16$^{+0}$ – 23$^{+6}$ weeks), *n*/total (%) | 18/758 (2.4) | 39/1,107 (3.5) | 57/1,865 (3.1) |
| Previous cervical surgery, *n*/total (%) | 22/758 (2.9) | 38/1,107 (3.4) | 60/1,865 (3.2) |
| Twin pregnancy, *n*/total (%) | 34/758 (4.5) | 40/1,107 (3.6) | 74/1,865 (4.0) |
| Known uterine anomaly, *n*/total (%) | 7/758 (0.9) | 4/1,107 (0.4) | 11/1,865 (0.6) |
| Cerclage, *n*/total (%) | 16/758 (2.1) | 21/1,107 (1.9) | 37/1,865 (2.0) |
| Progesterone, *n*/total (%) | 20/758 (2.6) | 9/1,104 (0.8) | 29/1,862 (1.6) |
| Arabin cervical pessary, *n*/total (%) | 0/758 (0.0) | 3/1,104 (0.3) | 3/1,862 (0.2) |
| Mean gestation at first visit in PTL, weeks (± SD) | 30.10 (3.25) *n* = 758 | 30.00 (3.28) *n* = 1,107 | 30.04 (3.27) |

NICU, neonatal intensive care unit; PTL, preterm labour; QUIPP, QUantitative Innovation in Predicting Preterm birth; sPTB, spontaneous preterm birth.

any site. In light of concerns around false-negative PTL predictive tests [11], all participants were reviewed who delivered <36 weeks' gestation and were not admitted and/or did not receive necessary ACSs (within 7 days of delivery) during the intervention period. Following intention-to-treat analysis, 4 women from the intervention sites and 12 from the control sites did not receive necessary management following one of their TPTL presentations (see S1 File).

## External validation of QUiPP app algorithms

During the trial, there were 2,845 visits with a qfFN on 2,430 women including 45 twin pregnancies. Visits with a CL in addition to qfFN were excluded, and women without a documented onset of labour and a gestation of delivery were excluded, leaving a validation set of 2,285 visits on 2,031 women. Iatrogenic births (excluding preterm prelabour rupture of membranes [PPROM]) within the specified time period were treated as missing because it was not reasonable to expect QUiPP to predict induced deliveries within the time period of interest

**Table 2. Impact of QUiPP app on the primary outcome (TPTL unnecessary management composite and individual components of composite) at intervention sites using QUiPP app versus control sites using conventional management.**

| Primary outcome | Intervention sites (724 women, 741 fetuses) | Control sites (1,075 women, 1,094 fetuses) | All (1,799 women, 1,835 fetuses) | OR | 95% CI | p-Value |
|---|---|---|---|---|---|---|
| **Composite primary outcome** | | | | | | |
| Unnecessary management | 84/741 (11.3%) | 126/1,094 (11.5%) | 210/1,835 (11.4%) | 0.97 | 0.66–1.42 | 0.883 |
| **Individual components of the primary outcome** | | | | | | |
| Unnecessary admissions (admitted and did not deliver <7 days) | 79/741 (10.7%) | 118/1,094 (10.8%) | 197/1,835 (10.7%) | 0.99 | 0.55–1.79 | 0.97 |
| Unnecessary discharges (discharged and did deliver <7 days) | 3/741 (0.4%) | 5/1,094 (0.5%) | 8/1,835 (0.4%) | 0.89 | 0.21–3.72 | 0.89 |
| Unnecessary IUT (IUT attempted and did not deliver <7 days) | 2/741 (0.3%) | 3/1,094 (0.3%) | 5/1,835 (0.3%) | 0.88 | 0.088–8.82 | 0.92 |
| EUT (transferred within 24 hours of delivery) | 0/741 (0.0%) | 2/1,094 (0.2%) | 2/1,835 (0.1%) | 1 | n/a | |

CI, confidence interval; EUT, ex utero transfer; IUT, in utero transfer; OR, odds ratio; QUIPP, QUantitative Innovation in Predicting Preterm birth; TPTL, threatened preterm labour.

(e.g., before 7, 14, or 28 days), but if they were induced outside this timeframe, the QUiPP prediction remained relevant. The mean gestation of visit was 30.3 (SD 3.23) weeks' gestation.

The receiver operating characteristic (ROC) curves (Fig 2) demonstrate the high predictive accuracy of the QUiPP app across the risk range: for preterm delivery within 7 days (as used in EQUIPTT) as well as within 14 and 28 days. The QUiPP app predicted PTB within 7 days with ROC 0.898 (0.850 to 0.946). This cohort also provides further validation for qfFN, with ROC of 0.902 (95% CI 0.857 to 0.946) for delivery within 7 days. As anticipated, there is little difference between the curves, as the qfFN value is mainly driving the QUiPP score in the low-risk cohort without CL measurements. However, the app also provides validated estimates of the individual probability of delivery within a fixed time.

**Table 3. Secondary outcomes of neonates at intervention sites using QUiPP app versus control sites using conventional management.**

| Secondary outcome measure | Neonates at intervention sites n/total (%) | Neonates at control sites n/total (%) | All neonates n/total (%) |
|---|---|---|---|
| Delivery within 7 days of presentation | 23/725 (3.2) | 40/1,069 (3.7) | 63/1,794 (3.5) |
| Delivery within 14 days | 43/725 (5.9) | 70/1,069 (6.5) | 113/1,794 (6.3) |
| Delivery within 28 days | 117/725 (16.1) | 167/1,069 (15.6) | 284/1,794 (15.8) |
| Preterm delivery <30 weeks | 9/725 (1.2) | 17/1,069 (1.6) | 26/1,794 (1.4) |
| Preterm delivery <34 weeks | 44/725 (6.1) | 53/1,069 (5.0) | 97/1,794 (5.4) |
| Preterm delivery <37 weeks | 127/725 (17.5) | 184/1,069 (17.2) | 311/1,794 (17.3) |
| Unnecessary ACSs | 63/725 (8.7) | 110/1,069 (10.3) | 173/1,794 (9.6) |
| Proportion of women <36/40 given necessary ACSs | 24/87 (27.6) | 24/117 (20.5) | |
| Unnecessary magnesium sulphate | 11/725 (1.5) | 9/1,069 (0.8) | 20/1,794 (1.1) |
| Proportion of women <30/40 given necessary magnesium sulphate | 3/9 (33.3) | 7/17 (41.2) | |
| Tocolysis at any visit | 17/718 (2.4) | 29/1,064 (2.7) | 46/1,782 (2.6) |
| Low birth weight (<2,500 g) | 118/723 (16.3) | 135/1,048 (12.9) | 253/1,771 (14.3) |
| Neonatal death | 2/725 (0.3) | 3/1,069 (0.3) | 5/1,794 (0.3) |
| Oxygen at 28 days | 7/725 (1.0) | 6/1,069 (0.6) | 13/1,794 (0.7) |

ACS, antenatal corticosteroid; QUIPP, QUantitative Innovation in Predicting Preterm birth.

**Table 4. Unnecessary admissions and discharges if 5% threshold was adhered to at intervention sites using QUiPP app versus control sites using conventional management.**

| Per protocol unnecessary management outcomes per visit | Intervention sites with QUiPP risks *n*/total (%) | Control sites *n*/total (%) | Both groups *n*/total (%) | AOR | 95% CI |
|---|---|---|---|---|---|
| Unnecessary admission (admitted and did not deliver <7 days) | 39/578 (6.7) | 129/1,351 (9.5) | 168/1,929 (8.7) | 0.623 | 0.349–1.11 |
| Unnecessary discharge (discharged and did deliver <7 days) | 4/578 (0.7) | 5/1,351 (0.4) | 9/1,929 (0.5) | 1.88 | 0.502–7.011 |
| Unnecessary admission or discharge | 43/578 (7.4) | 134/1,351 (9.9) | 177/1,929 (9.2) | 0.72 | 0.454–1.156 |

AOR, adjusted odds ratio; CI, confidence interval; QUIPP, QUantitative Innovation in Predicting Preterm birth.

## Discussion

### Main findings

The large sample of women with TPTL symptoms recruited in the EQUIPTT trial allowed extensive validation of the QUiPP app's predictive performance and captured rare insights into actual management of TPTL in a 24–7 setting. Using qfFN combined with individual risk factors and gestation, the excellent ROC values for prediction of sPTB within 7, 14, and 28 days support the use of the tool to triage TPTL. However, despite the accuracy of its prediction of sPTB, in the current study, the clinical introduction of the QUiPP app did not demonstrate a significant impact on the primary outcome: unnecessary management of TPTL. The lack of effect on primary outcome was likely to be related to the unexpected low event rate in the control arm, which was similar to that anticipated as a result of the intervention and related to clinicians in the control arm not following national guidance but incorporating aspects of the QUiPP app available to them, e.g., fetal fibronectin.

The use of CL in TPTL assessment was not frequent enough in these 13 hospitals to draw conclusions (it was only used in 5.5% of all visits). Contrary to NICE guidelines (which advises to not use predictive tests under 30 weeks' gestation), the majority of the control sites still used qfFN, the most significant component of the QUiPP risk. The control arm offers important evidence that the NICE recommended "treat-all strategy" at early gestations is being safely

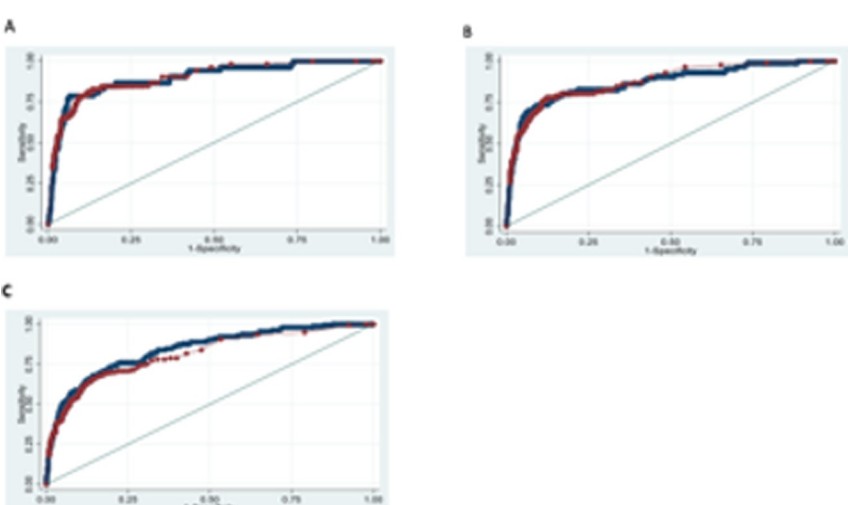

**Fig 2. ROC curves for QUiPP and qfFN prediction of sPTB within 7, 14, and 28 days using EQUIPTT participants as validation set.** A, B, and C show QUiPP qfFN (blue) and fetal fibronectin alone (red) ROC areas for preterm delivery within 7, 14, and 28 days of respectively. qfFN, quantitative fetal fibronectin; QUIPP, QUantitative Innovation in Predicting Preterm birth; ROC, receiver operating characteristic; sPTB, spontaneous preterm birth.

ignored in practice; there were no adverse events. The proportion of imminent deliveries in pregnancies receiving low risk scores (<5% within 7 days), representing "false-negative" assessments, was very low and corresponded to the app's predicted rates ($n = 6$) (i.e., it never gives a risk of 0%). These are not true false negatives, as this low rate is in keeping with the low risk predicted. As was advised in the EQUIPTT protocol, in the rare cases recorded in the study, clinical acumen or the mother's symptoms appropriately took precedence the low QUiPP scores and all the women received necessary and timely treatments for their preterm infants.

The external validation of the symptomatic qfFN algorithm of QUiPP confirmed its accuracy and generalisability for PTB prediction. Our findings are similar to preliminary reports of the Quantitative Fibronectin to help Decision-making in women with Symptoms of Preterm Labour (QUIDS) prospective study ($n = 2,924$, 26 sites 85 sPTB within 7 days ISRCTN41598423), which also provide comparable accuracy for prediction with qfFN alone compared to qfFN and CL (ROC 0.89), and these 2 independent studies have real potential to influence revisions in the national PTB guidelines [22].

Another important finding of this trial is how few mothers of preterm infants received necessary ACS, defined as administered within 7 days of delivery (23%), the most important antenatal intervention for preterm infants. QUiPP app use did appear to encourage necessary ACS administration (27.6% of infants <35 weeks received necessary ACS at intervention sites versus 20.5%), but this was not a powered outcome. While the impact on neonatal outcomes is not possible to measure from these data due to event rates, the need for timely ACS to reduce respiratory distress syndrome, intraventricular hemorrhage, and neonatal mortality is well established [23–25].

## Strengths and weaknesses

This cluster design allowed us to reach large numbers of women and clinicians across many centres in a short time period and with relative efficiency. Recruiting the entire group of women in TPTL, rather than a consented subset, allowed insights into actual use in pragmatic clinical settings rather than ideal conditions associated with traditional RCT design, enhancing the generalisability of our findings.

While total recruitment exceeded the target, there were disproportionately higher levels from the tertiary hospitals compared to smaller units with less established women's health academic departments. The 2 most research-active tertiary maternity units were both randomised to control. This typifies the relationship between engagement with research and high-quality clinical care [26], which, although beneficial for patients at research-active sites, can contribute to health inequalities across the National Health Service (NHS). Where there was less clinical or staffing capacity, reliance on pro formas over qfFN machine records may have led to selection bias and less representation of urgent and out-of-hours cases. Although statistical tests to allow for clustering did not alter the primary outcome, the variation in recruitment, TPTL management, and adoption of the QUiPP app between sites from the same region was profound. A larger number of centres from wider UK regions would be required to minimise this effect [27].

Our composite of unnecessary admissions, discharges, IUTs, and EUTs was a novel primary outcome aimed at measuring the true cost of either overly cautious (admitting too many women) or less careful practice (sending the wrong women home). Previous studies into the effectiveness of preterm prediction tests have chosen PTB rates as primary outcomes [28,29]. While PTB outcomes are easier to collect than management decisions, they are distant from the intervention and prone to influence by many confounding factors. However, the reasons

for admission are more complex than our methodology allowed for. For example, some women were assessed primarily for TPTL, but additional concerns prompted admission. Furthermore, reducing the primary outcome to a binary threshold (e.g., admit if risk of delivery >5% within 7 days), while necessary to measure impact, may not do justice to the QUIPP algorithms and grouping decisions around the 5% threshold classifies contrasting women together as low risk (e.g., 0.1% compared to 4.6%) or high risk (5.1% compared to 20%).

An alternative explanation of why clinical introduction of the QUiPP app was ineffective regarding the primary outcome is that we overstated the scale of routine unnecessary management of TPTL. Given that the power calculation was based on an unnecessary admission rate of 25% (reported in PETRA study, REC reference 14/LO/1988), and the unnecessary admission rate at control sites was actually 10.8%, it was not possible for QUiPP to demonstrate a significant reduction based mainly on admission rate. The randomisation of the most research-active tertiary centres to the control arm is likely to have distorted this degree of necessary management, as many were using qfFN and components of the app already; we could not mitigate against the app being used informally in the control arm. It may be that QUiPP has less added value in sites which already use qfFN judiciously, relative to those using the qualitative fFN 50 ng/mL cutoff to direct management or those with a treat-all strategy. The scrutiny of participating in a research trial itself could also have enhanced the performance of clinicians regardless of the randomised intervention [30]. The observed practice at control sites also suggests how far UK maternity services are deviating from NICE guidelines (to admit all women in TPTL prior to 30 week's gestation) [11] and supports the utility of qfFN and its place in the majority of local protocols.

The process outcomes of this trial, as well as women and clinician interviews (reported elsewhere), offered reassurance regarding the feasibility, acceptability, and fidelity of the QUiPP app in actual use. The per-protocol analysis suggests that closer adherence to QUiPP use and management guidance would have further improved TPTL management. Adoption and reach would have been enhanced by a more rigorous implementation strategy. It may not be the individual factors which hindered the ability of the innovation to deliver change but the dynamic interaction between them [31]. Our belief in the simplicity of the tool may have clouded recognition that QUiPP is a complex intervention involving multiple interacting components that may have been difficult for unfamiliar users to appreciate without a more focused implementation training package. Understanding the causal assumptions that underpin decisions to change practice are key to understanding QUiPP's mechanism to deliver change in the future.

## Conclusions

This cluster randomised trial did not demonstrate that the use of the QUiPP app reduced unnecessary management of TPTL compared to current management. However, the management of TPTL in the control arms was not according to national guidance and may not have reflected wider practice. The low rates of overtreatment or missed events in either arms support the interpretation of qfFN, with or without the QUiPP app, as a safe and accurate method for identifying women most likely to benefit from PTL interventions. It also highlights the need for greater consideration of implementation strategies for this type of research study.

## Supporting information

**S1 File. Table of clinical details of sPTBs within 7 days of TPTL presentation which were not appropriately managed during EQUIPTT trial.** sPTB, spontaneous preterm birth;

TPTL, threatened preterm labour.
(DOCX)

**S1 Table. CONSORT Statement for EQUIPTT trial.** CONSORT, Consolidated Standards of
Reporting Trials.
(DOCX)

**S1 CONSORT Checklist. CONSORT checklist for EQUIPTT Randomised.** CONSORT,
Consolidated Standards of Reporting Trials.
(DOCX)

**S1 Dataset. EQUIPTT baseline characteristics and risk factors for PTB.** PTB, preterm birth.
(XLSX)

**S2 Dataset. EQUIPTT outcomes.**
(XLSX)

**S3 Dataset. EQUIPTT ACSs outcomes.** ACS, antenatal corticosteroid.
(XLSX)

**S4 Dataset. EQUIPTT neonatal outcomes dataset.**
(XLSX)

**S1 Fig. Summary of relationships between main findings of EQUIPTT's evaluation of the
QUiPP app.** QUIPP, QUantitative Innovation in Predicting Preterm birth.
(PPTX)

## Acknowledgments

As it is a portfolio study, recruitment is supported through local clinical research networks.
HAW was also funded by a GSTT and KCL Biomedical Research Council Clinical Training
Fellowship Award. PTS is partly funded by Tommy's Charity and by NIHR Collaboration for
Leadership in Applied Health Research and Care, South London. JC was supported by an
NIHR/HEE CAT Clinical Doctoral Research Fellowship (CDRF-2013-04-026). The research
was supported by the NIHR Biomedical Research Centre based at GSTT and KCL and/or the
NIHR Clinical Research Facility.

   **Disclaimers:** The views expressed are those of the authors and not necessarily those of the
NHS, the NIHR, or the Department of Health and Social Care.

## Author Contributions

**Conceptualization:** Helena A. Watson, Paul T. Seed, Jenny Carter, Rachel M. Tribe, Andrew
   H. Shennan.

**Data curation:** Helena A. Watson, Naomi Carlisle, Paul T. Seed, Jenny Carter, Rachel M.
   Tribe, Andrew H. Shennan.

**Formal analysis:** Helena A. Watson, Naomi Carlisle, Paul T. Seed, Jenny Carter, Rachel M.
   Tribe.

**Funding acquisition:** Helena A. Watson, Rachel M. Tribe, Andrew H. Shennan.

**Investigation:** Helena A. Watson, Naomi Carlisle, Katy Kuhrt, Rachel M. Tribe, Andrew H.
   Shennan.

**Methodology:** Helena A. Watson, Naomi Carlisle, Paul T. Seed, Jenny Carter, Rachel M. Tribe, Andrew H. Shennan.

**Project administration:** Helena A. Watson, Naomi Carlisle, Katy Kuhrt, Andrew H. Shennan.

**Resources:** Helena A. Watson, Andrew H. Shennan.

**Supervision:** Rachel M. Tribe, Andrew H. Shennan.

**Validation:** Helena A. Watson, Paul T. Seed, Jenny Carter.

**Writing – original draft:** Helena A. Watson, Naomi Carlisle.

**Writing – review & editing:** Helena A. Watson, Naomi Carlisle, Paul T. Seed, Jenny Carter, Katy Kuhrt, Rachel M. Tribe, Andrew H. Shennan.

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
