## [Editor Report · Decision Letter 0]

14 Dec 2020

Dear Dr Watson, 

Thank you for submitting your manuscript entitled "EQUIPTT, a Cluster Randomised Trial for the Evaluation of the QUiPP app for Triage and Transfer in threatened preterm labour." for consideration by PLOS Medicine.

Your manuscript has now been evaluated by the PLOS Medicine editorial staff and I am writing to let you know that we would like to send your submission out for external assessment.

Once your full submission is complete, your paper will undergo a series of checks in preparation for full assessment.

Kind regards,

Richard Turner, PhD

Senior editor, PLOS Medicine

rturner@plos.org

---

## [Decision Letter · Decision Letter 1]

25 Jan 2021

Dear Dr. Watson,

Thank you very much for submitting your manuscript "EQUIPTT, a Cluster Randomised Trial for the Evaluation of the QUiPP app for Triage and Transfer in threatened preterm labour." (PMEDICINE-D-20-06019R1) for consideration at PLOS Medicine. 

Your paper was evaluated by a senior editor and discussed among all the editors here. It was also discussed with an academic editor with relevant expertise, and sent to four independent reviewers, including a statistical reviewer (r#2). The reviews are appended at the bottom of this email and any accompanying reviewer attachments can be seen via the link below:

[LINK]

In light of these reviews, I am afraid that we will not be able to accept the manuscript for publication in the journal in its current form, but we would like to consider a revised version that addresses the reviewers' and editors' comments. Obviously we cannot make any decision about publication until we have seen the revised manuscript and your response, and we plan to seek re-review by one or more of the reviewers. 

We expect to receive your revised manuscript by Feb 15 2021 11:59PM. Please email us (plosmedicine@plos.org) if you have any questions or concerns.

We look forward to receiving your revised manuscript. 

Sincerely,

Richard Turner, PhD, Senior Editor, 

PLOS Medicine

plosmedicine.org

*As noted by one reviewer - can the ISRCSN registry details for the trial (id - ISRCTN17846337; link - http://www.isrctn.com/ISRCTN17846337) be included within the submitted paper (ideally at end of abstract) - this is our usual style.

*The issue of individual informed consent (lack of) for this trial came up, and we recognise that the authors currently say in the paper that individual consent would not be feasible given the way the trial was conducted, and it's recognised for some cluster RCTs that individual consent can be waived. But it would be good as well to clarify (and state in the paper) whether the ethics committee specifically waived the need for informed consent.

*We'd suggest revising the title per the journal style, where these normally have a two-part format separated by colon (question/objective: study design) - example https://journals.plos.org/plosmedicine/article?id=10.1371/journal.pmed.1002949 (Implementing a structured model for osteoarthritis care in primary healthcare: A stepped-wedge cluster-randomised trial)

*At this stage please now restructure the abstract into the journal style - the subheadings for research papers should be Background, Methods and Findings, Conclusions (Methods and Findings is a single subhead). Within each subsection please use full sentences rather than (partial) sentence fragments. 

*In the last sentence of the Abstract Methods and Findings section, please include a note about any key limitation(s) of the study's methodology.

*We noticed a few typos/manuscript preparation errors, so please check through carefully before resubmitting the revision. Example in abstract - "Intervention sites were assigned to use the QUiPP app (with quantitative fetal fibronectin and/or or cervical length with history)in and suggested admission..." (the "in and suggested admission" part doesn't quite make sense. 

*Please refer to the CONSORT for cluster trials reporting guideline (https://www.equator-network.org/reporting-guidelines/consort-cluster/) and it might be helpful to add more detail on some aspects of reporting both to the paper (see reviewer 2's comments), and to the abstract, for example what the unit of randomization was (mat unit?) and how many were assigned to each arm. For submitted trials the journal normally wants to see a completed CONSORT checklist appended as supporting information - so we'd recommend adding this (and the CONSORT for cluster RCTs would probably be the most helpful here).

*Some minor changes to the writeup would also be helpful, we'd suggest moving the "Aims and Objectives" subsection into Methods and also moving the Ethics Approval information from the end of the paper to the end of the Methods.

*At this stage, we ask that you include a short, non-technical Author Summary of your research to make findings accessible to a wide audience that includes both scientists and non-scientists. The Author Summary should immediately follow the Abstract in your revised manuscript. This text is subject to editorial change and should be distinct from the scientific abstract. Please see our author guidelines for more information: https://journals.plos.org/plosmedicine/s/revising-your-manuscript#loc-author-summary.

Comments from the reviewers:

Reviewer #1: It is a relevant study as presently, AI based algorithms are being used increasingly in medical practice to improve decision making. 

Reviewer #2: Statistical review

This paper reports a cluster randomised trial comparing use of an app to guide decision about whether to admit pregnant women at risk of pre-term birth. There were no significant differences between the arms, which appears to be due to control sites using qfFN, which is a major contributor to the app risk score.

I did not see a protocol provided, or details of whether the trial was registered on a repository such as clinicaltrials.gov, which is required for PLOS medicine. My other comments were more minor:

1. Abstract - I found it hard to follow line 15 and 16 of the abstract, in particular the part outside the brackets.

2. Abstract, main outcome measures - I think it's a bit unclear here whether the outcomes described here are separate primary outcomes or combined into one. Later in the paper it's clear it's the latter but here I was confused by 'main outcome measures' as the sub-title. 

3. Abstract, main outcome measures - it's also not too clear whether the outcome was binary, i.e. occurrence of one or both of the events, or as written, the sum of number of events. I would recommend this is made clearer. A similar comment applies to the methods section on page 7.

4. Abstract - for the primary outcome I would recommend a p-value is included as well as the confidence interval.

5. Page 5 - can more information be provided about the randomisation procedure and how it was performed?

6. Page 8 - the statistical analysis says that risk ratios were calculated but the abstract reports odds ratios. This section also mentions 'minimisation process' which should be described more in the randomisation section as per previous comment.

7. Page 8 - I would recommend providing more details of the statistical analysis for evaluating the predictive accuracy of the QUiPP app.

8. Figure 1 - it would be of interest to know whether the loss-to-follow-up was similar between the two groups. Maybe this could be given by arm in the figure?

9. Page 11 - the authors alternate between using 'cluster' period and 'analysis' period which I found a bit confusing - are these the same things?

10. Table 3 - were the neonatal outcomes not planned to be analysed statistically? If so then I'd provide the results as in table 2.

11. Page 13 - I found the per-protocol analysis difficult to follow: could this be described more in the methods section?

James Wason

Reviewer #3: 

This cluster trial offers a deep insight on the clinical management of threatened PTL, in UK.

The planning and execution of the study have been well performed.

Overall, a 17% PTB rate in women presenting (admitted or not) for PTL are within expected range in settings of good quality of care, not being affected by excessive fear and anxiety.

The trial demonstrated that clinicians behave better than guidelines provided for them! Certainly, app would warrant a more standardized application of risk factors.

I congratulate Authors for this very interesting study.

Few points to better understand data:

1) There should be an explanation why lot of tertiary centres pertain to control. It was expected that standard of cares could be better than in other units. Thus we believe app will be less added value?

2) How many centres used "old" yes/not fFN test? In the case, is possible a comparison between quali- and quanti-tative test on main outcome?

3) How was steroids rate administered after 34 weeks? 

Reviewer #4: 

The objective of this multi-centre cluster randomised-controlled trial was to investigate the impact of the use of the QUiPP app versus conventional management on inappropriate management for threatened preterm labour (TPTL). According to the intention to treat analysis, inappropriate management of TPTL was 11.3% at the intervention sites versus 11.5% at control sites (OR 0.97; 95% CI 0.66-1.42). According to the per protocol analysis, inappropriate admissions and discharges were also not significantly reduced in the intervention arm (7.4% versus 9.9%).

This is an interesting randomised-controlled trial which should be considered for publication. However, I have several comments.

The methodology of this RCT is adequate with a low risk of bias:

Internal validity:

- A selection bias is very unlikely: EQUIPTT was a cluster randomised-controlled trial with a parallel group design across 13 obstetric centres: randomisation was performed at the cluster (maternity centre) level.

- Because masking was not feasible, performance bias is possible but unlikely because the QUiPP app was only implemented in the intervention centres and management of women was similar between the two arms.

- A detection bias is very unlikely because the composite primary outcome of inappropriate management for TPTL was objective and defined by the number of inappropriate admission decisions: admitted and delivery interval >7 days or not admitted and delivery interval ≤ 7 days, and the number of inappropriate in-utero transfer decisions/actions: IUTs that occurred or were attempted >7 days prior to delivery, and ex utero transfers that should have been in-utero.

- An attrition bias is possible but also unlikely: indeed, 1872 women (761 intervention and 1111 control) were recruited but only 1799 women (724 intervention and 1075 control) were included in the ITT analysis.

External validity : 

13 obstetric centres including Special Care Unit (level 1), local Neonatal Unit (level 2) and Neonatal Intensive Care Unit (level 3) participated in this study, ensuring an excellent external validity.

However, this paper presents several limitations that should be considered:

The per protocol analysis is interesting to report. However, this analysis is not specified in the Methods section (Statistical Analysis). The authors should clarify whether this is a prespecified analysis or a post-hoc analysis.

A main strong of this RCT is its pragmatic approach and the absence of significant statistical differences between the intervention and control arms, both in ITT analysis and per protocol analysis reinforces the reliability of the results. Therefore, the post-hoc analysis evaluating the impact of the QUiPP app compared with current UK national guidance is not consistent with the pragmatic approach of the authors and should be deleted. This post-hoc analysis could even be considered a "spin" leading the authors to consider the usefulness of the QUiPP app while their trial is negative and shows the lack of benefit to the use of the QUiPP app in real life.

Therefore, The Results and the Discussion Sections should be reviewed, mainly focusing on the prespecified analysis reporting negative results. I understand that the authors are disappointed by this negative result but this "spin" has the effect of devaluing the methodological quality of this trial.

Patrick Rozenberg

[LINK]

---

## [Decision Letter · Decision Letter 2]

11 Apr 2021

Dear Dr. Watson,

Thank you very much for re-submitting your manuscript "Evaluating the use of the QUiPP app and its impact on the management of threatened preterm labour: a cluster-randomised trial." (PMEDICINE-D-20-06019R2) for consideration at PLOS Medicine.

I have discussed the paper with editorial colleagues and it was also seen again by two reviewers. I am pleased to tell you that, provided the remaining editorial and production issues are dealt with, we expect to be able to accept the paper for publication in the journal.

[LINK]

Please let me know if you have any questions, and we look forward to receiving the revised manuscript shortly.   

Sincerely,

Richard Turner, PhD

rturner@plos.org

Requests from Editors:

Please quote study dates in your abstract. 

Please quote aggregate demographic details for study participants in the abstract.

Around line 17 of the abstract, please quote the number of women and hospitals in each group, and specify "randomly assigned". 

Where you quote the primary endpoint findings in your abstract, please add absolute numbers (e.g., "84/741 fetuses (11.3%) versus ...").

After quoting the primary endpoint findings in your abstract, we suggest adding a sentence to quote findings of the prespecified per-protocol analysis.

Around line 31 of the abstract, please add a sentence, say, to quote the absolute numbers of "adverse events" (i.e., 4 vs 12 women not receiving necessary management). 

A "close bracket" appears to be missing at line 33 of the abstract. 

The final sentence of the "Methods and findings" subsection of your abstract should begin "Study limitations include ..." or similar and quote 2-3 of the study's main limitations. Please either add a new sentence to achieve this or adapt the current relevant sentence in your abstract.

Please ensure that "TPTL" is spelt out at first use (for example, it appears that "(TPTL)" needs to be added preceding the full point at line 41). 

Please also spell out "qfFN" at first use and use the abbreviation consistently. 

Please adapt the author summary to consist of three subsections, individually consisting of three short bulleted points each. You may find it helpful to consult one or two recent research papers in PLOS Medicine to get a sense of the preferred style. 

Please make that "data were" on p.7 of the PDF.

Throughout the text, please style the reference call-outs as follows: "... prediction studies [18,19]." (noting the absence of spaces within the square brackets); and adopt journal style (e.g., "[23-25]" in the Discussion). 

Please remove the information on funding from the end of the main text. In the event of publication, this information will appear in the article metadata, via entries in the submission form. 

In the reference list, please convert all italics to plain text. 

Noting reference 5, please ensure that all references have full access details. 

Noting reference 31, please ensure that where appropriate 6 author names are listed, followed by "et al.".

Please rename the attached CONSORT checklist "S1_CONSORT_Checklist" and refer to it by this label in your Methods section. Please adapt the checklist so that individual items are referred to by section (e.g., "Methods") and paragraph number rather than by line or page numbers, as the latter generally change upon publication. 

Comments from Reviewers:

*** Reviewer #2: 

Thank you to the authors for responding to my previous comments well. I have no remaining issues to raise.

*** Reviewer #4: 

The modifications made by the authors are satisfactory.

The article is now suitable for publication.

***

[LINK]

---

## [Editor Report · Decision Letter 3]

8 Jun 2021

Dear Dr Watson, 

On behalf of my colleagues, I am pleased to inform you that we have agreed to publish your manuscript "Evaluating the use of the QUiPP app and its impact on the management of threatened preterm labour: a cluster-randomised trial." (PMEDICINE-D-20-06019R3) in PLOS Medicine.

Prior to final acceptance, please: correct "ISRCTN" on the title page; check the quoted number of women in the intervention arm in the abstract ("762", quoted as "761" in the results, we think); make that "30.2 years" in the abstract; move the first point in the second subsection of the author summary to the first subsection; and add full access details to reference 6.

PRESS

Sincerely, 

Richard Turner, PhD 

rturner@plos.org